

# The impact of self-incompatibility systems on the prevention of biparental inbreeding

Tara N. Furstenau[1,2] and Reed A. Cartwright[1]

[1] The Biodesign Institute and School of Life Sciences, Arizona State University, Tempe, AZ, United States of America
[2] The School of Informatics, Computing, and Cyber Systems, Northern Arizona University, Flagstaff, AZ, United States of America

Corresponding author
Reed A. Cartwright,
cartwright@asu.edu

## ABSTRACT

Inbreeding in hermaphroditic plants can occur through two different mechanisms: biparental inbreeding, when a plant mates with a related individual, or self-fertilization, when a plant mates with itself. To avoid inbreeding, many hermaphroditic plants have evolved self-incompatibility (SI) systems which prevent or limit self-fertilization. One particular SI system—homomorphic SI—can also reduce biparental inbreeding. Homomorphic SI is found in many angiosperm species, and it is often assumed that the additional benefit of reduced biparental inbreeding may be a factor in the success of this SI system. To test this assumption, we developed a spatially-explicit, individual-based simulation of plant populations that displayed three different types of homomorphic SI. We measured the total level of inbreeding avoidance by comparing each population to a self-compatible population (NSI), and we measured biparental inbreeding avoidance by comparing to a population of self-incompatible plants that were free to mate with any other individual (PSI). Because biparental inbreeding is more common when offspring dispersal is limited, we examined the levels of biparental inbreeding over a range of dispersal distances. We also tested whether the introduction of inbreeding depression affected the level of biparental inbreeding avoidance. We found that there was a statistically significant decrease in autozygosity in each of the homomorphic SI populations compared to the PSI population and, as expected, this was more pronounced when seed and pollen dispersal was limited. However, levels of homozygosity and inbreeding depression were not reduced. At low dispersal, homomorphic SI populations also suffered reduced female fecundity and had smaller census population sizes. Overall, our simulations showed that the homomorphic SI systems had little impact on the amount of biparental inbreeding in the population especially when compared to the overall reduction in inbreeding compared to the NSI population. With further study, this observation may have important consequences for research into the origin and evolution of homomorphic self-incompatibility systems.

## INTRODUCTION

A large portion of angiosperm species (∼72%) produce hermaphroditic flowers (*Yampolsky & Yampolsky, 1922*), which bear both male (stamen) and female (carpel) reproductive systems. In some cases, these species are capable of self-fertilizing and reproducing without a mating partner. Self-fertilizing plants benefit from reproductive assurance when mates are limited (*Darwin, 1876*; *Baker, 1955*; *Stebbins, 1957*; *Busch, 2005*; *Herlihy & Eckert, 2005*) and have a pollen transmission advantage over outcrossing plants (*Fisher, 1941*). Nevertheless, outcrossing remains the dominant reproductive strategy in angiosperms (*Igic & Kohn, 2006*), likely due to the negative effects of inbreeding depression outweighing the advantages of self-fertilization (*Darwin, 1876*; *Charlesworth & Charlesworth, 1979b*). Many hermaphroditic plant species exhibit a wide variety of morphologically- or molecularly-enforced self-incompatibility (SI) systems to avoid self-fertilization.

In morphology-enforced or heteromorphic SI systems, self-fertilization is reduced through spatial or temporal separation of the male and female reproductive organs (anther and stigma, respectively). For example, *Darwin (1862)* first described the heterostyly SI system in *Priumula* (*P. vulgaris* and *veris*), in which each plant expresses one of two flower morphologies that differ in the relative heights of the anther and stigma. The different arrangements ensure that pollinating insects that visit the anther of one morph will only deposit pollen on stigmas with the opposite morph.

Molecularly-enforced or homomorphic SI systems are more common and are found in species spanning at least 100 angiosperm families (*Igic, Lande & Kohn, 2008*). In homomorphic plants, the female reproductive system is able to recognize and reject self-generated pollen using various molecular mechanisms. In order for self-recognition to be successful, the genes controlling the molecular phenotypes of the pollen and the carpel must be inherited together. Typically, these phenotypes are controlled by two genes at the *S* locus that are tightly linked, due to repressed recombination (*Casselman et al., 2000*; *Castric et al., 2010*; *Charlesworth & Awadalla, 1998*; *Kamau, Charlesworth & Charlesworth, 2007*; *Kawabe et al., 2006*; *Vieira, Charlesworth & Vieira, 2003*), and highly polymorphic. The *S* locus is under negative, frequency-dependent selection and pollen with rare *S* phenotypes are favored. Homomorphic SI mechanisms ensure that plants will recognize and reject all self-generated pollen as well as some pollen from closely related plants. As a result, homomorphic SI systems not only reduce inbreeding by preventing self-fertilization, they also reduce mating between close relatives (biparental inbreeding *Charlesworth & Charlesworth, 1987*).

It is often assumed that the success of homomorphic SI systems across the angiosperms is due to this two-fold inbreeding avoidance strategy. Unfortunately, because the genetic outcomes of biparental inbreeding and self-fertilization are similar, it is difficult to distinguish between these two types of inbreeding in natural populations without a controlled experimental setup (*Griffin & Eckert, 2003*). This makes it difficult to draw strong conclusions about the role of biparental inbreeding avoidance in the evolution of homomorphic SI systems.

Mixed-mating models and genetic markers are often used to estimate levels of biparental inbreeding (*Ennos & Clegg, 1982*), but these estimates can be inaccurate even when a large number of loci are used (*Ritland, 2002*). These estimates are complicated by the fact that mixed-mating models assume that a certain proportion of progeny are a product of self-fertilization while the rest are a product of outcrossing with random unrelated individuals. In many natural plant populations, however, outcrossing is more likely to occur with related individuals, particularly when the population displays fine-scale spatial genetic structure (*Zhao, Xia & Lu, 2009*). Because angiosperms are sessile, dispersal is achieved through the movement of pollen and seed, and in many species, pollen and seed dispersal distances rarely exceed more than a few meters (*Fenster, 1991*; *Levin, 1981*). Consequently, plants will often become established near their parents surrounded by related individuals, which they will likely mate with. This phenomenon, known as isolation-by-distance, results in greater genetic similarity between individuals that are near each other than those that are farther away. The increased potential for biparental inbreeding under isolation-by-distance can be beneficial because it reduces the genetic cost of outrcrossing by increasing parent-offspring relatedness (*Uyenoyama, 1986*). On the other hand, it may also increase homozygosity allowing the expression of deleterious recessive alleles.

The extent to which inbreeding is detrimental depends on the history of inbreeding in the population. Both biparental inbreeding and self-fertilization can increase homozygosity within a genome, and inbred offspring are more likely to express recessive deleterious alleles and suffer reduced viability and fecundity (*Charlesworth & Charlesworth, 1987*; *Charlesworth, Morgan & Charlesworth, 1990*). While, self-fertilization facilitates purifying selection to purge highly deleterious alleles, a large number of slightly deleterious alleles can be maintained (*Charlesworth, Morgan & Charlesworth, 1990*; *Wang et al., 1999*). Outcrossing species tend to maintain recessive deleterious alleles in a heterozygous state, which can lead to inbreeding depression. When biparental inbreeding is common, some of the segregating deleterious alleles can be purged in outcrossing populations. However, in many plant species, crosses between close neighbors have been shown to produce less fit offspring, and because the reduction in fitness is associated with spatial proximity, this is likely evidence of inbreeding depression resulting from isolation-by-distance (*Heywood, 1991*).

Previous studies have provided evidence that homozygosity is reduced in regions of the genome that are linked to the *S* locus. The forced heterozygosity at the *S* locus extends to other linked loci and can reduce the expression of recessive deleterious alleles at those loci. Deleterious alleles can accumulate in this region because they are sheltered from selection (*Llaurens, Gonthier & Billiard, 2009*). It remains unclear, however, whether homomorphic SI systems reduce biparental inbreeding at loci that are not linked to the *S* locus. *Cartwright (2009)* presented results from a simulation study which showed a large decrease in biparental inbreeding in homomorphic SI simulations near the *S* locus, but at unlinked loci, the reduction in biparental inbreeding was relatively small. This suggests that at unlinked loci, homomorphic SI systems may only have a small impact on the amount of biparental inbreeding. This study, however, did not model inbreeding depression which may provide a selective advantage to avoid inbreeding.

In this current study, we analyzed the impact homomorphic SI systems have on the level of biparental inbreeding in a population. We used a spatially-explicit, individual-based simulation to model plant populations with three different homomorphic SI systems. To differentiate between inbreeding due to self-fertilization and biparental inbreeding we compared the level of inbreeding (autozygosity and homozygosity) in the homomorphic SI populations to a partial SI (PSI) population where individuals were obligate out-crossers but no genetic SI system was in place to prevent biparental inbreeding. Any decrease in the level of inbreeding in the SI populations compared to the PSI population would be due to biparental inbreeding avoidance. To determine the reduction in total inbreeding, the populations were also compared to a self-compatible population that had no SI system (NSI). We focused on measuring the level of inbreeding at loci that were not linked to the *S* locus, and we incorporated inbreeding depression by simulating the segregation of recessive deleterious alleles in the population.

The three homomorphic SI systems vary in the way that they discriminated against pollen from plants with a matching *S* allele, and they are each described in the Fig. 1 diagram. The first system was modeled after the gametophytic SI system (GSI), which is the most widespread SI system and is found in Solanaceae, Rosaceae and Scrophulariaceae (*Franklin-Tong & Franklin, 2003*). In GSI systems, the pollen phenotype is solely determined by the *S* haplotype that it inherits. From a single diploid plant, roughly 50% of the pollen will carry one *S* haplotype, and 50% will carry the other *S* haplotype. If two plants have one *S* allele in common, half of the pollen from each plant—those that do not carry the common haplotype—will be able to fertilize the other plant.

The second system is modeled after the sporophytic SI system that is common in Brassicaceae (BSI). One often studied example is *Arabidopsis lyrata*, a self-incompatible relative of the self-compatible model angiosperm, *Arabidopsis thaliana* (*Kusaba et al., 2001*; *Charlesworth et al., 2003*; *Mable, Schierup & Charlesworth, 2003*; *Kawabe et al., 2006*; *Kamau, Charlesworth & Charlesworth, 2007*; *Schierup, Bechsgaard & Christiansen, 2008*). In the BSI system, the phenotype of the pollen is determined by the diploid *S* genotype of the parent plant. Dominance relationships exist between the *S* alleles and the pollen expresses the phenotype of the dominant allele. If two plants share the same dominant *S* allele, they will be unable to interbreed; however, if they share only the same recessive *S* allele, all of the pollen will be compatible between the two plants. Consequently, this is the only system that potentially allows a plant to become homozygous for recessive *S* alleles (*Hiscock & Tabah, 2003*).

Finally, we modeled a sporophytic SI (SSI) system that is similar to BSI except all the *S* alleles are codominant. There is no known biological equivalent of this SI system, and a situation where all *S* alleles are equally codominant is highly unlikely. Nevertheless, the SSI system serves to model an extreme case of discrimination where pollen is prevented from fertilizing any plant that shares either *S* allele. We predict that this more stringent SI system will show the greatest reduction in biparental inbreeding. In each of the homomorphic SI systems, we treat the *S* alleles that control the female phenotype as codominant.

Because biparental inbreeding is more likely when dispersal is limited, we ran each simulation with a range of different seed and pollen dispersal distance parameters. We

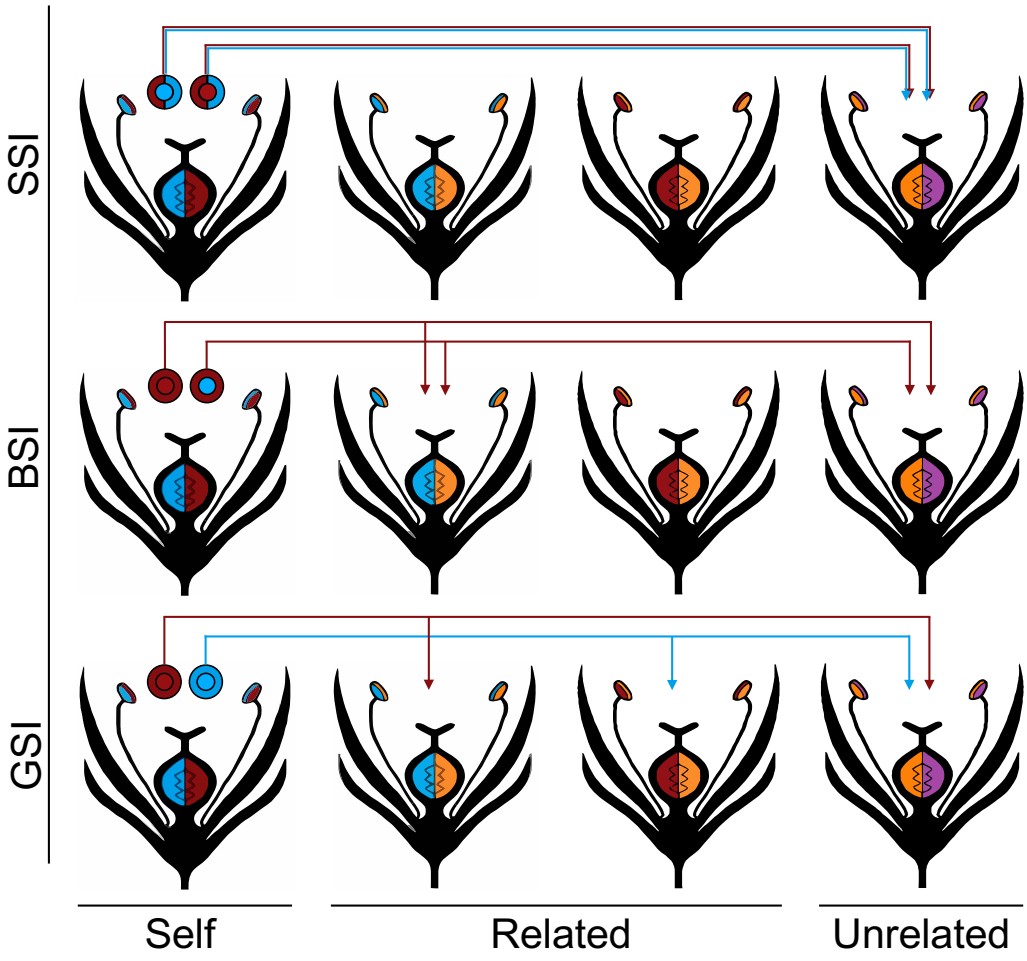

**Figure 1** **Homomorphic self-incompatibility systems.** Four different *S* haplotypes are represented by the colors red, blue, orange, and purple. The plants in the first column produce the pollen represented by the circles above each plant. The color of the inner circle indicates the pollen's haplotype and the outer circle indicates the pollen's phenotype in each of the three SI systems: GSI, BSI, and SSI. Under the GSI system, the pollen phenotype is the same as the pollen haplotype; under the BSI system, the red allele is dominant to the blue allele so all of the pollen are phenotypically red; and under the SSI system, both of the parental alleles are codominant so both are expressed in the pollen phenotype. The *S* alleles controlling the female phenotype are all codominant. In every SI system, none of the pollen is compatible with the plant that produced it (self) and all of the pollen is compatible with unrelated individuals that do not share any of the same *S* alleles with the parent plant. The arrows indicate which related plants (those that share one *S* allele with the parent plant) are compatible with each of the pollen types.

suspected that the homomorphic SI populations would have the greatest advantage over the PSI population, in terms of biparental inbreeding avoidance, when the dispersal distances were small. We also tracked the population size and the number of viable seeds produced to monitor population declines and female fecundity.

## METHODS

### Simulation

We developed a spatially-explicit, individual-based simulation to model discrete generations of self-incompatible plant populations. In the simulation, populations inhabited a $100 \times 100$ toroidal lattice (cf. *Epperson, 1995*; *Robledo-Arnuncio & Rousset, 2010*; *Rousset, 1997*; *Slatkin & Maddison, 1990*; *Slatkin, 1993*) where each cell was occupied by a single, hermaphroditic individual. The plants were diploid and had several, independently assorting genetic loci.

In plants, the *S* locus consists of multiple, tightly linked, highly polymorphic genes; however, in our simulations we treated the *S* locus as a single gene, with multiple alleles. Typically, the formation of novel functional *S* haplotypes through mutation is rare because it requires coordination between the genes controlling both the pollen and the carpel phenotypes; a mutation in just one component will result in the breakdown of self-incompatibility (*Charlesworth & Charlesworth, 1979a*; *Uyenoyama, Zhang & Newbigin, 2001*; *Igic, Lande & Kohn, 2008*). Therefore, in the simulation, we kept the mutation rate at the *S* locus low ($\mu_s = 10^{-5}$) and each mutation resulted in either a completely new *S* haplotype according to the infinite alleles mutation model (*Kimura & Crow, 1964*) or changed to one of 10 alleles according to the K-alleles mutation model (*Crow & Kimura, 1970*) where specified. We did not allow mutations that would result in the breakdown of SI.

The marker locus, *M*, was used to measure the amount of inbreeding in the population. The alleles at the *M* locus were all selectively neutral and mutated at rate $\mu_m = 10^{-4}$, under the infinite alleles model. The higher mutation rate at the *M* locus maintained higher levels of polymorphism which aided in the estimation of inbreeding. In the initial population, each *S* and *M* allele was unique and the simulation was run for a 10,000-generation burn-in period to reach a drift-mutation equilibrium.

Each individual carried a total of 10 independent deleterious loci ($D_1, D_2, \ldots, D_{10}$) that were not linked to each other or to any other locus. Each *D* locus carried either a wild-type allele or a recessive deleterious allele. In the initial population, all individuals carried wild-type alleles that permanently mutated into deleterious alleles at rate $\mu_d = 0.1$; this resulted in a genome-wide recessive mutation rate that was close to 1. Each homozygous recessive genotype at a *D* locus increased the probability that an individual would be sterile by 0.005. Individually, these alleles were only slightly deleterious and thus were more likely to be maintained in the population; in combination, they produced an appreciable number of sterile individuals. Affected individuals were viable but were unable to produce pollen or seed. Typically, the probability that a deleterious mutation occurs at a single locus is rare, but the probability of a deleterious mutation occurring across the whole genome is high. Therefore, to maintain a large enough penalty for inbreeding, we used a high mutation rate at each *D* locus so that, on average, there would be one new deleterious mutation per haploid gamete.

At the beginning of each generation, fertile parent plants produced gametes—10 pollen grains and five ovules—through the independent assortment of loci. Pollen grains were dispersed from the parent's location according to a normal distribution along each axis

with standard deviation $\sigma$. Incoming pollen was checked for compatibility with the plant in the new location based on the rules of the designated SI system. If compatible, the pollen was randomly assigned to an ovule; otherwise it was discarded. When pollen dispersal was complete, some ovules remained unfertilized while other ovules had a pool of pollen from which one pollen grain was randomly chosen. Unfertilized ovules were aborted and fertilized ovules formed seeds. Seeds were then dispersed from the parent's location in the same way as the pollen. When seed dispersal was complete, a single seed from each cell was randomly selected to become a parent in the next generation. Mutations occurred in the germ line of the parents before they produce gametes so all of their offspring carried the mutation. The pollen and seed dispersal parameters were both set to either $\sigma = 1, 2, 4,$ or 6.

In each simulation, pollen compatibility was determined by one of the five different compatibility systems: NSI, PSI, GSI, BSI, and SSI. A serial dominance scheme, similar to that described in *Vekemans, Schierup & Christiansen (1998)*, was used to model the dominance relationships between the *S* alleles in the BSI system. The *S* alleles were sorted into a dominance hierarchy such that each allele was dominant to all alleles below it and recessive to all alleles above it in the hierarchy; new alleles, introduced through mutations, were randomly inserted into the hierarchy. For the self-compatible NSI system, outcrossing occurred when pollen dispersed outside of the parent cell; otherwise, self-fertilization occurred. Consequently, self-fertilization was relatively more common than outcrossing when pollen dispersal distance was limited.

Source code for the simulation is available at https://github.com/tfursten/SI-cpp/ (doi: 10.5281/zenodo.1016153).

## Analysis

A random sample of 500 individuals was collected from the population every 10,000 generations for a total of 500 nearly independent samples. To measure inbreeding in each sample, we calculated the proportion of sampled individuals that were autozygous and homozygous at the *M* locus. An individual's *M* alleles were considered to be autozygous (identical-by-descent) if they both descended from the same allele in a grandparent (autozygosity through the parents is not possible for SI systems), regardless of mutation. Individuals were homozygous if their two *M* alleles were the same.

To analyze the results, we used the Anderson-Darling two-sample test (*Scholz & Stephens, 1987*) implemented in the kSamples R package (*R Core Team, 2015*; *Scholz & Zhu, 2016*). The test statistic was $T_a = (AD - (k-1))/\sigma$, and the *p*-value estimation method was set to simulate the default 10,000 random rank permutations using the average rank score for ties. The distribution of values (proportion of heterozygotes and autozygotes) was compared between the different simulations under the null hypothesis that the values came from the same underlying distribution. The *p*-values from the pairwise comparisons were adjusted for multiple tests using the Holm correction (*Holm, 1979*), and the significance criterion was set at 0.05 for all tests.

We also recorded the average number of alleles at the *S* and *M* locus, and the average squared parent-offspring dispersal distance ($s^2$). In plants, $s^2 = \sigma_s^2 + \sigma_p^2/2$, where $\sigma_s$ represents seed movement and $\sigma_p$ represents pollen movement (*Crawford, 1984*). In this

formula, seed dispersal contributes more than pollen dispersal because seeds carry gametes from both parents whereas pollen only carries gametes from the father. From the whole population, we recorded the total number of adults, the number of seeds produced, and the number of sterile individuals.

## RESULTS

### Effect of inbreeding depression

Unlike the simulations in *Cartwright (2009)*, our simulations included inbreeding depression which should provide a selective advantage to avoid inbreeding. To determine if the simulated inbreeding depression had an effect, we compared simulations with (Del) and without (Neu) a penalty for inbreeding (Fig. 2). Introducing inbreeding depression resulted in a significant difference in the level of homozygosity. Median homozygosity was lower in each SI system when inbreeding depression was present. Comparing across the different SI systems, PSI, GSI, BSI, and SSI were not significantly different from each other in simulations with and without inbreeding depression (See Table S1 for a list of *p*-values). Autozygosity, which measures very recent inbreeding, was not significantly different in simulations with and without inbreeding depression within the same SI system, except for in the PSI system. In simulations with and without inbreeding depression, the NSI systems had significantly higher level of autozygosity and the SSI system had a significantly lower level of autozygosity.

### Reduction in biparental inbreeding

The amount of inbreeding in the PSI system was used as a baseline value to determine how much biparental-inbreeding-avoidance occurred in the homomorphic SI simulations. Because the PSI system only prevented self-fertilization, any reduction in inbreeding below the level observed in the PSI simulations represented a reduction in biparental inbreeding. Compared to the large drop in inbreeding between the NSI and PSI systems (due to the prevention of self-fertilization), the decrease in inbreeding between PSI and the homomorphic SI systems was statistically significant but relatively small. Figure 3 shows the empirical density plot of measures of autozygosity in each of the simulations, and the difference between the homomorphic SI systems and the PSI system was about an order of magnitude smaller than the difference between PSI and NSI. Autozygosity was not significantly different between the GSI and BSI systems (inset), but it was significantly lower in the more stringent SSI system.

### Isolation-by-distance

Biparental inbreeding is more common under isolation-by-distance and the biparental inbreeding avoidance strategy provided by homomorphic SI systems may provide a greater advantage in this situation. To test this, we compared the amount of inbreeding in simulations with various pollen and seed dispersal distance parameters. Figure 4 shows pairwise comparisons of the level of homozygosity and autozygosity in each SI system with a range of dispersal distance parameters. Homozygosity was significantly higher when isolation-by-distance was strongest but there was no difference between the different SI

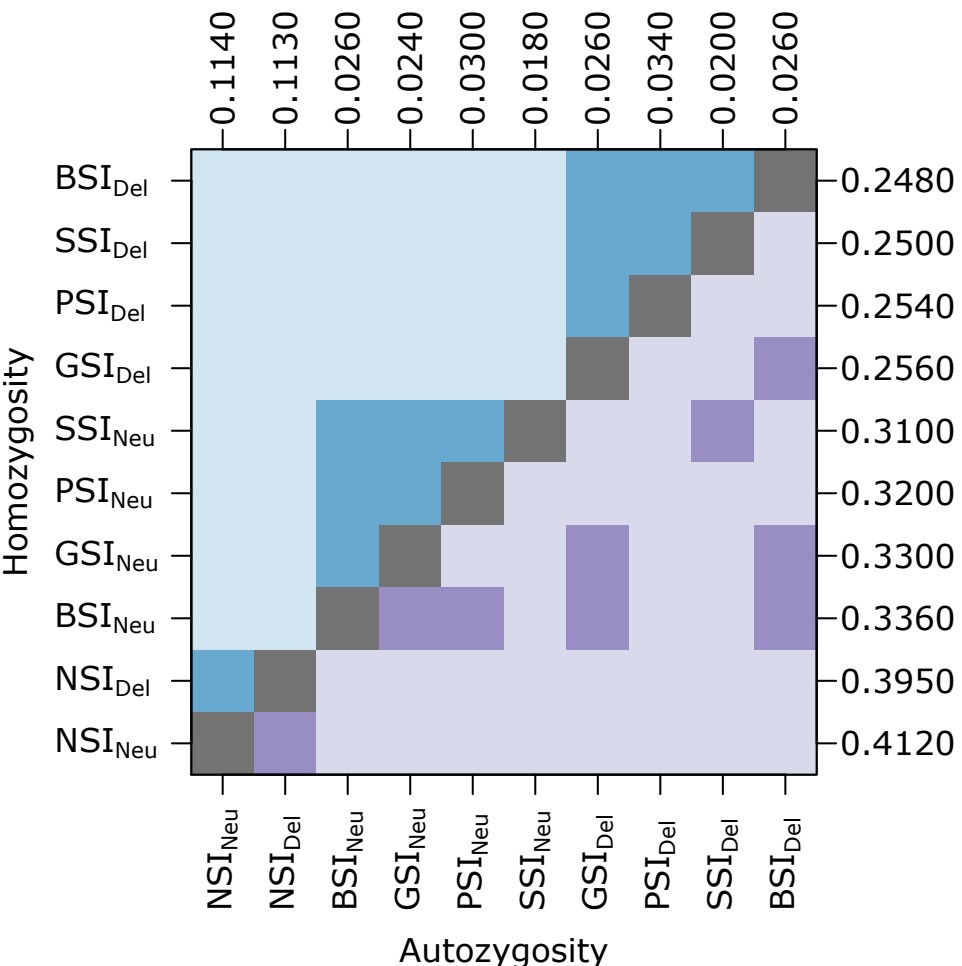

**Figure 2** **In the homomorphic SI simulations, inbreeding depression had a significant effect on homozygosity.** The plot shows pairwise comparisons for each SI system with inbreeding depression (Del) or without inbreeding depression (Neu). The upper (blue) and lower (purple) triangles compare the distribution of proportions of homozygotes and autozygotes, respectively, in 500 samples from each simulation. The color of each square indicates the outcome of a Holm-corrected, pairwise Anderson-Darling test which determined whether the pair of distributions were significantly different ($p < 0.05$; light color) or not (dark color). The values along the right and top axes are the median homozygosity and autozygosity for each simulation, respectively. The simulations are sorted on both axes by median homozygosity. The simulations were run on a $100 \times 100$ landscape with pollen and seed dispersal parameter $\sigma = 1$.

systems. When isolation-by-distance was strongest ($\sigma = 1$), autozygosity was significantly different between all of the SI systems except BSI and GSI. The median autozygosity was highest for PSI and lowest for SSI. When isolation-by-distance was weak ($\sigma = 6$), autozygosity was not significantly different between BSI, GSI, and SSI but each of these were still significantly different from PSI. The median autozygosity at $\sigma = 6$ was 0 for each SI system and the average autozygosity was 0.0011 for PSI, 0.0007 for GSI, 0.0008 for BSI, and 0.0006 for SSI. Overall, we observed a greater decrease in the median autozygosity

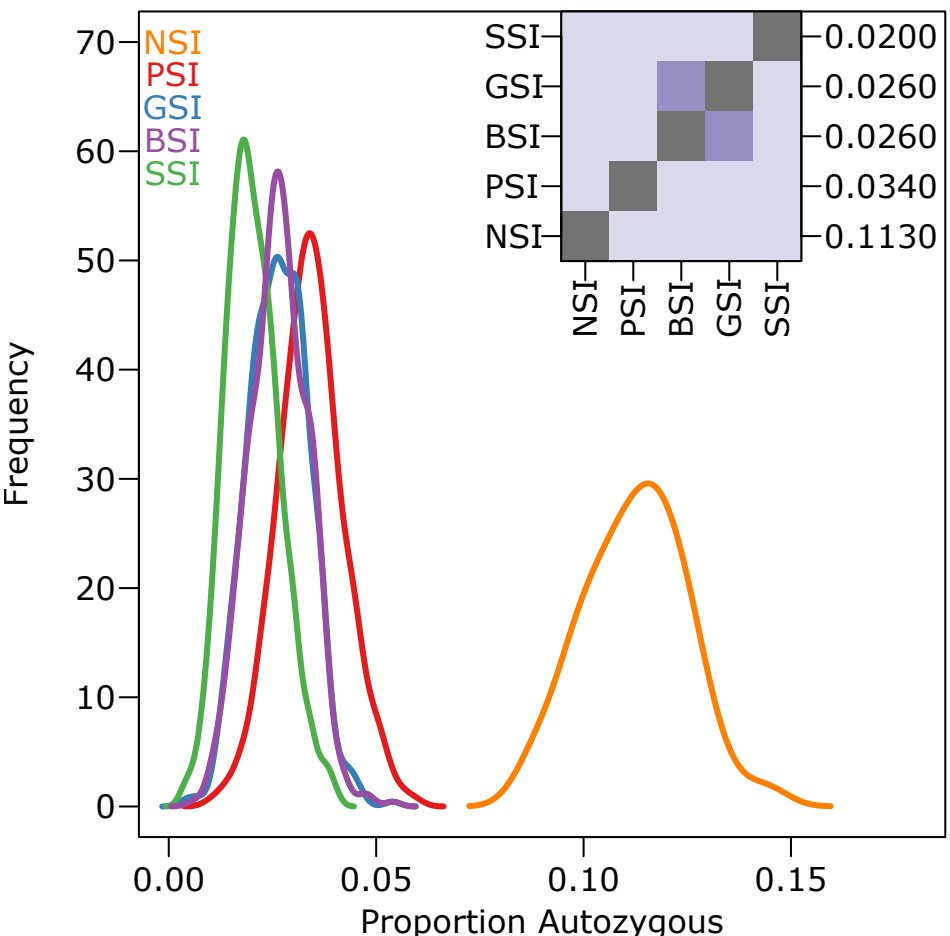

**Figure 3 The reduction in biparental inbreeding in homomorphic SI systems was small compared to the reduction in self-fertilization.** The empirical density plot (main) shows the distribution of the proportion of autozygotes in 500 samples from simulations of each SI system. The inset shows the pairwise comparisons of each distribution where the color of each square indicates the outcome of a Holm-corrected, pairwise Anderson–Darling test which determined whether the pair of distributions were significantly different ($p < 0.05$; light color) or not (dark color). The values along the right axis of the inset are the medians in increasing order. The simulations included inbreeding depression and were run on a 100 $\times$100 landscape with pollen and seed dispersal parameter $\sigma = 1$.

levels, in both absolute and relative differences, between PSI and the homomorphic SI simulations when isolation-by-distance was stronger.

## Population demographics and allele diversity

Tracking population parameters in the simulation allowed us to better understand how the SI systems effect population size, inbreeding depression, fecundity, effective dispersal, and allele diversity. Table 1 provides a summary of median per-generation population demographic values for each of the simulations including: census population size, the number of sterile individuals, seed set, dispersal distance, and the number of alleles at the $M$ locus and the $S$ locus.

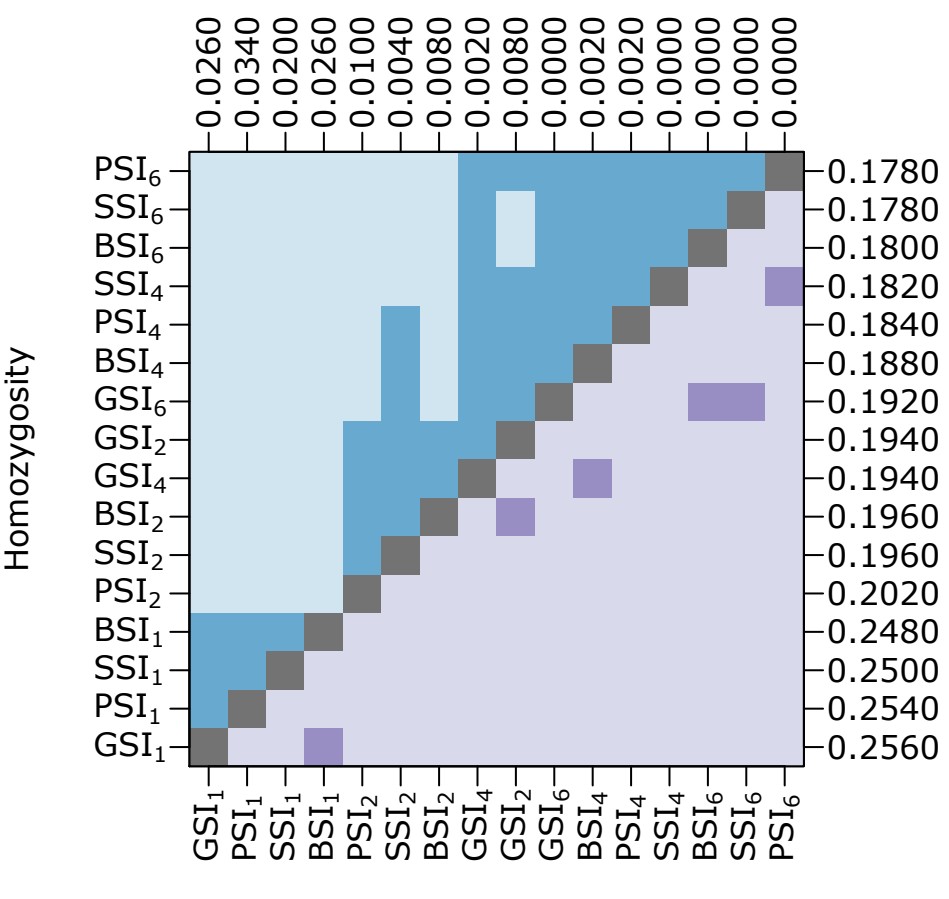

**Figure 4 Biparental inbreeding avoidance in homomorphic SI systems was greater when isolation-by-distance was stronger.** The plot shows pairwise comparisons for simulations of each SI system at different dispersal levels. The upper (blue) and lower (purple) triangles compare the distribution of proportions of homozygotes and autozygotes, respectively, in 500 samples from each simulation. The color of each square indicates the outcome of a Holm-corrected, pairwise Anderson–Darling test which determined whether the pair of distributions were significantly different ($p < 0.05$; light color) or not (dark color). The values along the right and top axes are the medians for homozygosity and autozygosity, respectively. The simulations are sorted on both axes by median homozygosity. The simulations included inbreeding depression and were run on a $100 \times 100$ landscape with pollen and seed dispersal parameters $\sigma = 1, 2, 4$, and 6, indicated in the subscript.

In the simulations, the maximum allowable population size was 10,000 individuals; however, in many cases the population size was smaller because seeds would fail to disperse into some locations, especially when fewer seeds were produced. The PSI simulations had the largest median population size and the population size was not affected by different dispersal distance parameters. In the homomorphic SI simulations, the number of individuals increased with dispersal distance. When $\sigma = 1$, the homomorphic SI simulations all had a significantly reduced population size and the greatest reduction was observed under the SSI system.

**Table 1  Seed set and population size is reduced when the SI system is more stringent.** The table provides the medians for the number of individuals ($N$), the number of sterile individuals, the seed set, the mean squared parent-offspring dispersal distance ($s^2$), the number of unique $M$ alleles, and the number of unique $S$ alleles for simulations with different SI systems and different dispersal parameters ($\sigma$). The maximum possible number of individuals in the population is 10,000 and the maximum number of seeds is 50,000.

|     | $\sigma$ | $N$ | Sterile | Seed set | $s^2$ | $M$ alleles | $S$ alleles |
|-----|------|-------|---------|----------|-------|-------------|-------------|
| PSI | 1 | 9,907 | 486 | 45,679 | 1.72 | 24 | 4 |
|     | 2 | 9,905 | 485 | 46,351 | 6.20 | 23 | 4 |
|     | 4 | 9,905 | 486 | 46,458 | 24.24 | 22 | 4 |
|     | 6 | 9,905 | 485 | 46,481 | 54.11 | 22 | 4 |
| GSI | 1 | 9,886 | 484 | 44,127 | 1.72 | 23 | 75 |
|     | 2 | 9,902 | 486 | 45,963 | 6.21 | 22 | 73 |
|     | 4 | 9,903 | 482 | 46,296 | 24.27 | 22 | 73 |
|     | 6 | 9,903 | 484 | 46,340 | 54.06 | 22 | 73 |
| BSI | 1 | 9,879 | 484 | 43,547 | 1.73 | 24 | 50 |
|     | 2 | 9,900 | 485 | 45,755 | 6.21 | 22 | 38 |
|     | 4 | 9,902 | 484 | 46,107 | 24.22 | 22 | 35 |
|     | 6 | 9,901 | 484 | 46,166 | 54.16 | 22 | 34 |
| SSI | 1 | 9,847 | 482 | 41,555 | 1.73 | 24 | 81 |
|     | 2 | 9,897 | 484 | 45,447 | 6.24 | 23 | 76 |
|     | 4 | 9,901 | 485 | 46,072 | 24.18 | 22 | 75 |
|     | 6 | 9,901 | 484 | 46,174 | 54.10 | 22 | 75 |

Inbreeding in the population increased the probability that sterile individuals were produced. The average number of sterile individuals per generation across all simulations was 484.5 which represents approximately 5% of the population. For most of the dispersal levels, the PSI simulations had the highest number of sterile individuals; although, none of the differences were statistically significant.

A maximum of 50,000 seeds can be produced in one generation (10,000 individuals with five ovules each). Seed set was highest in the PSI simulations and lowest in the SSI simulations. In each of the SI systems, seed set increased as dispersal distance increased.

The expected mean-squared parent-offspring dispersal distances were 1.5, 6, 24, and 54 for dispersal parameters 1, 2, 4, and 6, respectively. The observed $s^2$ values were slightly higher across all simulations but the relative difference was much greater when isolation-by-distance was strong. The $s^2$ values were not significantly different between the different SI systems when $\sigma = 2$, 4, and 6, but when $\sigma = 1$, the SSI simulation had significantly higher effective dispersal than GSI and PSI.

In the homomorphic SI systems, high diversity is maintained at the $S$ locus. The SSI system maintained the largest number of $S$ alleles followed by the GSI system then the BSI system. Few alleles were maintained at the $S$ locus in the PSI system because the $S$ allele was not active, essentially behaving as a selectively neutral marker. The number of alleles maintained at both the $S$ locus and the $M$ locus decreased as the average dispersal distance increased in all of the SI systems.

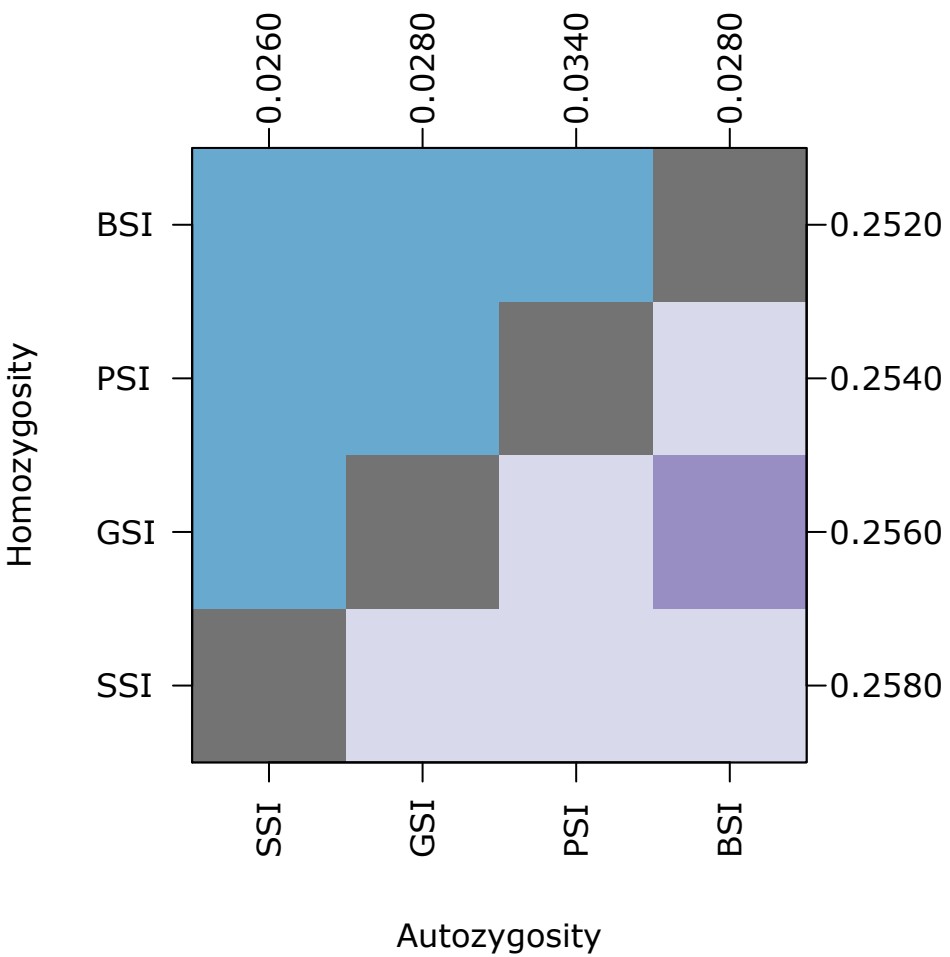

**Figure 5** **Biparental inbreeding avoidance is fairly stable with respect to S allele diversity.** The plot shows pairwise comparisons for simulations of each SI system in which the number of *S* alleles were reduced to 10 under the k-alleles mutation model. The upper (blue) and lower (purple) triangles compare the distribution of proportions of homozygotes and autozygotes, respectively, in 500 samples from each simulation. The color of each square indicates the outcome of a Holm-corrected, pairwise Anderson–Darling test which determined whether the pair of distributions were significantly different ($p < 0.05$; light color) or not (dark color). The values along the right and top axes are the medians for homozygosity and autozygosity, respectively. The simulations are sorted on both axes by median homozygosity. The simulations included inbreeding depression and were run on a $100 \times 100$ landscape with $\sigma = 1$ for the pollen and seed dispersal parameters.

## Reduced S-allele diversity

At equilibrium, the homomorphic SI populations maintained a high number of *S* alleles. Because the number of *S* alleles in natural populations is often lower, we ran simulations where the maximum number of *S* alleles was ten (under the k-alleles mutation model). Homozygosity and autozygosity were largely unaffected in these simulations (Fig. 5). Similar to the simulations with greater S-diversity, homozygosity was not significantly different for any of the SI systems. Autozygosity was significantly different between all SI

systems except for GSI and BSI. The median autozygosity and homozygosity increased slightly for SSI and BSI compared to the simulations with more *S* allele diversity.

## DISCUSSION

Measuring the amount of inbreeding in the PSI simulations was important because it allowed us to tease apart inbreeding due to self-fertilization and inbreeding between related individuals in the homomorphic SI simulations. The statistically significant decrease in autozygosity in the homomorphic SI systems compared to the PSI system supports the assumption that homomorphic SI reduces biparental inbreeding. These results also suggest that biparental inbreeding avoidance is stronger under isolation-by-distance, when seed and pollen dispersal is limited. Nevertheless, the reduction in inbreeding due to biparental inbreeding avoidance was negligable, even under strong isolation-by-distance, compared to selfing avoidance. This suggests that the effect may not be biologically significant. While these results were from simulations with a population size of 10,000 individuals, we repeated these simulations with a range of population sizes (2,500, 40,000, and 160,000) and verified that the pattern we observed was consistent (*Furstenau, 2016*).

Our results agree with *Cartwright (2009)*, which found that autozygosity at loci that are unlinked to the *S* locus is only slightly lower in homomorphic SI compared to PSI. That study, however, did not include inbreeding depression which would have provide a selective advantage for plants that avoided inbreeding. We found that when a penalty for inbreeding was introduced, there was a significant reduction in the proportion of homozygotes at the unlinked locus but the proportions were not different between the homomorphic SI and PSI systems. The penalty did not affect autozygosity except in the case of the PSI population which increased. This may have been a consequence of the type of inbreeding penalty that we introduced. Seeds that were impacted by the deleterious effects of inbreeding were viable but they were not fertile so they effectively took up space and reduced the number of potential mates for neighboring plants. The reduced mating pool near these individuals may have increased the potential for biparental inbreeding particularly in the PSI system where there was no genetic mechanism to avoid it. We have also explored modeling inbreeding depression in different ways including using rare deleterious alleles with more extreme fitness effects and an early acting effect which resulted in aborted ovules rather than sterile offspring. The simulations that we ran using both of these conditions were indistinguishable from the simulations without inbreeding depression. We settled on the final model because it provided a strong enough penalty that could be detected (on average about 5% of the population was sterile) and the deleterious mutations were not quickly purged from the population.

Among the different homomorphic SI systems, the BSI and GSI outcomes were not significantly different in most cases. The simple linear dominance scheme that we used to model the relationships between the *S* alleles in the BSI system is likely responsible for the similarities between the two systems. Under the GSI model, if we consider three related plants with *S* genotypes $S_1S_2$, $S_2S_3$, and $S_1S_3$, the first plant would be able to accept approximately 50% of the pollen produced by both plants two and three –the

pollen with the $S_3$ haplotype in both cases. Under the BSI system with linear dominance ($S_1 > S_2 > S_3$), the first plant can accept 100% of the pollen from the second plant but it would not be compatible with the third plant. In both systems, the first plant receives the same total amount of pollen from the related plants, the only difference is that the number of compatible mates is higher under the GSI system. In many cases in Brassicaceae, the dominance between $S$ alleles is not linear; for example, the $S$ alleles in self-incompatible field mustard (*Brassica campestis*) and marrow-stem kale (*Brassica oleracea*) fall into general classes that are dominant and recessive to each other while alleles within the same group are codominant (*Bateman, 1955*; *Thompson, 1957*; *Thompson & Taylor, 1966*; *Hatakeyama et al., 1998*). Under these more complicated dominance patterns, the $S$ allele frequency dynamics may cause very different behavior in the BSI system.

In homomorphic SI systems, the $S$ locus is under negative frequency-dependent selection which favors low frequency alleles (*Wright, 1939*). Self-incompatible plant populations tend to maintain a high level of $S$ allele polymorphism; however; when $S$ allele diversity is reduced (e.g., due to a bottleneck or population fragmentation) potential mates become scarce which can lead to population declines or a breakdown of the SI system (*Byers & Meagher, 1992*; *Leducq et al., 2010*; *Reinartz & Les, 1994*; *Young & Pickup, 2010*). We found that, at equilibrium, the SSI simulations maintained the highest number of $S$ alleles, especially when dispersal was restricted. Because the SSI system was the most strict—plants were only compatible when they had completely different $S$ alleles—there was a selective advantage to having a unique set of $S$ alleles that was different than neighboring individuals. The BSI populations maintained the fewest $S$ alleles and this was likely due to a weakening of the negative frequency dependent selection for the recessive $S$ alleles (*Billiard, Castric & Vekemans, 2007*; *Schierup, Vekemans & Christiansen, 1997*; *Vekemans, Schierup & Christiansen, 1998*). The number of $S$ alleles maintained in the population was very high compared to what would normally be expected in a natural population. *Mable, Schierup & Charlesworth (2003)* estimated that there were 25 $S$ alleles in a population of *Arabidopsis lyrata* and according to *Lawrence (2000)*, the largest number of $S$ alleles that have been identified in a species is 49. To ensure that the low level of biparental inbreeding avoidance that we observed was not an artifact of the large number of $S$ alleles maintained in the population, we ran additional simulations with a maximum of 10 $S$ alleles. We found that the number of $S$ alleles did not have an impact on our results.

Homomorphic SI systems have a negative effect on population size and female fecundity (*Vekemans, Schierup & Christiansen, 1998*). Fecundity selection in the simulation was modeled by limiting the number of pollen grains produced by each plant. After pollen dispersal, each plant had a finite pollen pool that was further reduced when a high proportion of the pollen grains are incompatible. If the number of compatible pollen grains were less than the number of ovules, there was a reduction in seed set. The lowest seed set was observed in the SSI simulations because it had the strictest rules for compatibility. Seed set was lowest when dispersal was limited because the pollen pool consisted of a higher proportion of close neighbors which were more likely to be related and thus incompatible. This also resulted in higher effective dispersal distances in the SSI populations. The reduction in seed set translated into a reduction in the census population size which

then further limited the number of available mates in the next generation. Seed set and population size were significantly higher for the PSI populations at each dispersal level, which suggests that reduced fecundity and population size was unique to the homomorphic SI systems. Smaller populations are not able to maintain high levels of $S$ diversity which reduces the number of compatible mates and ultimately reduces seed set. As a result, population size continues to decline and the population is likely to go extinct. This raises concerns for endangered SI species suffering from habitat fragmentation and population bottlenecks such as *Arnica montana*, a grassland perennial in Europe (*Luijten et al., 2000*); *Aster furcatus* (Forked aster) of the midwestern United States (*Les, Reinartz & Esselman, 1991*); three cliff dwelling species, *Sonchus pustulatus*, *S. fragilis*, and *S. masguindalii*, of the western Mediterranean Basin (*Silva, Brennan & Mejías, 2016*); and *Hymenoxys acaulis var. glabra* (grassland daisy) of the Great Lakes region (*Demauro, 1993*).

## ACKNOWLEDGEMENTS

The authors would like to thank J Malukiewicz, MS Rosenberg, R Schwartz, J Taylor, M Wilson-Sayres, D Winter, and S Wu for helpful comments, and K Dai for programing tips.

### Funding
The authors received no funding for this work.

### Competing Interests
Reed A. Cartwright is an Academic Editor for PeerJ.

### Author Contributions
- Tara N. Furstenau conceived and designed the experiments, performed the experiments, analyzed the data, contributed reagents/materials/analysis tools, wrote the paper, prepared figures and/or tables, reviewed drafts of the paper.
- Reed A. Cartwright conceived and designed the experiments, wrote the paper, prepared figures and/or tables, reviewed drafts of the paper.

### Data Availability
Simulation source code is available at https://github.com/tfursten/SI-cpp and archived at https://zenodo.org/record/1016153 (doi: 10.5281/zenodo.1016153).

### Supplemental Information
Supplemental information for this article can be found online at http://dx.doi.org/10.7717/peerj.4085#supplemental-information.

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
