# Peer review of "The impact of self-incompatibility systems on the prevention of biparental inbreeding"

_PeerJ, doi:10.7717/peerj.4085_

## Round 0.1 · original submission · Major Revisions

· Academic Editor

Major Revisions

Two talented reviewers have read your paper, and while both think it warrants publication in PeerJ, both also have several concerns that will need to be addressed. In particular I urge you to pay special attention to reviewer 1's questions about whether you are testing the hypothesis you claim and reviewer 2's concerns about the clarity and the realism of the SI systems being simulated.

Reviewer 1 ·

Basic reporting

On the whole, the paper is well written, and the background literature is well reviewed. The figures are clear and well presented.

That said, there are times where the literature is incorrectly explained, and the relationship between the major hypothesis and the results is not particularly solid (see below).

Another major shortcoming of the manuscript is that the computer code is not provided. It should be uploaded with the next submission, and must be stored permanently on dryad conditional on acceptance.

However, all of these concerns can be addressed relatively easily.

Experimental design

I believe the research is well executed and suitable for this journal. As said in '1. Basic reporting' it is not repeatable without computer code, but I get the sense of their approach.

The investigation is rigorous and honest.

Validity of the findings

I believe that the work as stated was well executed.

A concern I had is that results were described as significant or insignificant without the use of p-values. I would suggest a different approach (if computationally possible) - that the authors increase their sample size such that differences are apparent without the need to invoke null hypothesis significance testing. If that degree of replication is not feasible, than the authors should provide p-values with their statistical statements.

My most pressing concern is the relationship between the motivating question and the results. I describe this below.

Additional comments

This is an interesting paper. That said I have a significant concern about the framing of this paper with regard to the major motivating hypothesis.

1. The authors claim to investigate if differences in the extent of biparental inbreeding associated with alternative incompatibility systems can favor chemical based self-rejection, and claim to reject this hypothesis. However, the simulation does not address this problem. Rather, the authors investigate differences in the genetic load and the extent of biparental inbreeding as a function of alternative self-incompatibility systems, and never directly test their proposed hypothesis. I therefore would like to see the authors ask directly if different modes of self-incompatibility change the extent of autozygosity, allozygosity, and inbreeding depression (the question they asked). The question of selection for this form of self-rejection should be relegated to speculation in the discussion.

2. In fact it is not clear what such a test of this hypothesis would be. The authors do not make clear wether they imagine such incompatibility systems replacing one another in natural populations, or if they expect higher level species selection to act. This later hypothesis is often invoked to explain the distribution of plant mating systems (e.g. Goldberg et al 2010). These levels of selection should be clarified in the introduction (lines 34-38).

3. I am somewhat worried about the sensitivity of results to a few somewhat idiosyncratic decisions concerning the nature of the genetic load that was simulated (modest selection for a small number of loci with high mutation rates) and the expression of this load (individuals with high inbreeding depression take up space in the lattice). I would be curious to see results with more, rarer deleterious recessives with more extreme fitness effects (as we generally assume). I would also be interested to see how early-acting inbreeding depression (such that unfit inbreds did not take up space) would impact the results.

4. There are a few (relatively minor) misstatements in the introduction. As the authors' know being that 95% of plants are hermaphrodites does not mean "that a single plant is capable of self-fertilizing and reproducing without a mating partner," but rather that plants have male and female parts. The authors should clarify this for their audience. I also think that Fisher's automatic advantage could be explained better. The advantage is that selfers can fertilize themselves and someone else not that they pass a higher fraction of their genes to their offspring (if all pollen that self-fertilized an individual fertilized a different individual instead, fitness would be identical). Another minor point is that heteromorphic incompatibility systems often have genetic based rejections, however, I don't think this would impact the modelling or the results.

Reviewer 2 ·

Basic reporting

This manuscript is nicely written, with professional language and clear figures. Ideas and relevant literature frame the context well (except for the specific concerns noted below). There are no data to be archived, but the code is available.

Experimental design

This is a primary research manuscript with a clear motivating question. Investigation of the question is rigorous (except for the specific concerns noted below). The code provided and settings described are enough to replicate the research.

Validity of the findings

The findings are summarized appropriately. Conclusions are mostly stated clearly, though some improvements are noted below. My concerns about the overall biological interpretation are noted below.

Additional comments

This manuscript is focused on the general question of how different systems of self-incompatibility affect biparental inbreeding. This is an interesting question to ask for both evolutionary and ecological reasons. The authors have done an excellent job of clearly and concisely presenting the results from many simulations. Before this work is published, however, it is essential to clear up some issues about the interpretation of the SI systems and conclusions.

(1) My one big concern is about "heteromorphic SI." The model implements a system called PSI, in which selfing is prohibited but there are no additional mating incompatibilities. This system is useful as a control, in that it does nothing to reduce biparental inbreeding. However, it bears no resemblance to real heteromorphic SI systems, in which intra-morph matings (not only selfs) are prohibited by an S-locus that is linked to one or more morphological loci. The authors only admit that PSI is an entirely artificial system near the end (lines 375-378), and the entire introductory framing, as well as much of the discussion, is cast as heteromorphic versus homomorphic SI. This is extremely confusing and misleading. Please either completely remove the discussion of heteromorphic SI, or simulate new results using a genetic system that reflects how heteromorphic SI works in nature.

(2) My other main concern is about the central conclusion: that homomorphic SI systems do not do much to prevent biparental inbreeding. For one thing, some of the statements about angiosperm evolution are far too sweeping to be based on this one small simulation study (e.g., lines 27-28, 322-323). Additionally, I suspect this finding from the model may in part be an artifact of the large number of S-alleles (Table 1). I think these are at least twice as large as what's known from natural systems, which would greatly reduce the potential to avoid biparental inbreeding. It would actually be quite interesting to see how the inbreeding avoidance scales with the number of S-alleles, at least a comparison between "few" (say, 10) and "many" (say, 40). Alternatively, the rate of generating new S-alleles could be lowered. Either way, please provide some references to support that the number of S-alleles in the model is biologically realistic.

(3) My remaining comments are much smaller, ordered by appearance rather than importance.

throughout: "biparental" is more usual than "bi-parental"

throughout: "Mating system" typically refers to the selfing rate rather than the type of SI. But admittedly the literature is frustratingly inconsistent with this kind of terminology.

throughout: SI does not always act in the stigma. For most or all GSI, it acts in the style.

30: Renner (2014) says that 5-6% of species are dioecious, but there are lots of other sexual systems (monoecy, gynodioecy, etc.) besides hermaphroditic.

30-31: Just because an individual is hermaphroditic doesn't mean it is capable of uniparental reproduction (e.g., SI).

35: The logic of "consequently" is not clear.

52-54: Using "haplotype" here means it isn't correct for sporophytic SI.

67-68: Can an empirical citation be provided?

81: Purging hasn't been explained.

91: It's not clear what "continuous" means here.

115: "if they share *only* the same recessive"

129-143: I think of inbreeding as defined at the level of which *individuals* mate with which, so it sounds a bit odd to say that inbreeding is reduced at particular parts of the genome. Maybe homozygosity would be clearer?

148: The statements here are true (but see a comment below), but "for this reason" is not their logical connection.

158: For isolation by distance, a toroid doesn't seem appropriate. Better to allow dispersal off the edges?

209-211: Why is autozygosity defined by grandparents rather than by parents? If the latter, it seems like selfing would automatically be separated from biparental inbreeding. I'm wondering if the current "proportion autozygous" results (Fig 3) might be dominanted by selfing, obscuring differences between the SI systems.

220-227: This would make more sense if there were some indication of what quantities will be compared.

Results: It would help if each subsection began with an indication of what is being checked or what questions are being answered, rather than just jumping into descriptive results. Lines 259-262 do a good job of this.

The Discussion leads off with a small matter that is not the main point and sounds like an artifact. It would be more effective to start with big questions and perspective.

313-314: I don't follow this sentence. Several generations from when?

341, 346, 348, and 148-149: The claim is that there are many S-alleles in a population *because* that's what's necessary for enough compatible matings. This reads like population-level group selection in the sense that real populations or simulation runs without enough S-alleles preferentially went extinct. Is this the intent? Natural SI populations with few S-alleles do exist, e.g., after a bottleneck.

373-375: This seems like an extremely aggressive claim with no foundation in logic or data.

386-389: SI can also weaken in older flowers simply because proteins denature, rather than because it's a beneficial strategy.

Table 1: The s^2 column doesn't seem necessary, or it could at least be placed right next to the sigma column.

Figure 1: This is a nice depiction. One small adjustment is that the pollen's inner red circle looks orange (on my screen, at least).

---

## Round 0.2 · Minor Revisions

· Academic Editor

Minor Revisions

Thank you for your changes to the manuscript and careful attention to reviewer comments. I think these look good, but would request you add in mention of the citations to other work that have used tori for isolation by distance (as mentioned in the response to review).

---

## Round 0.3 · accepted · Accept

· Academic Editor

Accept

Thanks for the quick turnaround.